# Molecular Immune-Inflammatory Connections between Dietary Fats and Atherosclerotic Cardiovascular Disease: Which Translation into Clinics?

**DOI:** 10.3390/nu13113768

**Published:** 2021-10-25

**Authors:** Elisa Mattavelli, Alberico Luigi Catapano, Andrea Baragetti

**Affiliations:** 1Department of Pharmacological and Biomolecular Sciences, University of Milan, 20133 Milan, Italy; elisa.mattavelli@unimi.it (E.M.); alberico.catapano@unimi.it (A.L.C.); 2S.I.S.A. Centre for the Study of Atherosclerosis, Bassini Hospital, Cinisello Balsamo, Cinisello Balsamo, 20092 Milan, Italy; 3IRCCS Multimedica Hospital, Sesto San Giovanni, 20092 Milan, Italy

**Keywords:** dietary lipids, immune-inflammation, cardiovascular disease, microbiota

## Abstract

Current guidelines recommend reducing the daily intake of dietary fats for the prevention of ischemic cardiovascular diseases (CVDs). Avoiding saturated fats while increasing the intake of mono- or polyunsaturated fatty acids has been for long time the cornerstone of dietary approaches in cardiovascular prevention, mainly due to the metabolic effects of these molecules. However, recently, this approach has been critically revised. The experimental evidence, in fact, supports the concept that the pro- or anti-inflammatory potential of different dietary fats contributes to atherogenic or anti-atherogenic cellular and molecular processes beyond (or in addition to) their metabolic effects. All these aspects are hardly translatable into clinics when trying to find connections between the pro-/anti-inflammatory potential of dietary lipids and their effects on CVD outcomes. Interventional trials, although providing stronger potential for causal inference, are typically small sample-sized, and they have short follow-up, noncompliance, and high attrition rates. Besides, observational studies are confounded by a number of variables and the quantification of dietary intakes is far from optimal. A better understanding of the anatomic and physiological barriers for the absorption and the players involved in the metabolism of dietary lipids (e.g., gut microbiota) might be an alternative strategy in the attempt to provide a first step towards a personalized dietary approach in CVD prevention.

## 1. Introduction

The favorable transition from *hominids* to *homo sapiens* during evolution [1] prompted changes in the physiological functions and immune competences (including survival to pathogens and infections) to adapt to the intake of high-energy containing foods, principally dietary fats [2]. Today, the access to a variety of highly-caloric fatty foods is hardly balanced by energy consumption. As a consequence, the dominant genetic pathways evolved to favor the intake of calorie-rich diets and the storage of energy as fats in the adipose tissue are in some circumstances redundant, especially in affluent societies, giving rise to obesity, diabetes and cardiovascular disease (CVD)-comorbidities.

Currently, the Western lifestyle, including dietary habits, is believed to contribute a chronic state of low-grade inflammation [3] that eventually prompts the development of atherosclerosis, the etiopathological factor of ischemic CVDs. Of note is that the connection between the Western dietary lifestyle and onset of CVDs has been demonstrated to impact morbidity and mortality worldwide [4].

Dietary interventions are considered the first approach in preventing atherosclerotic CVDs. All guidelines, while recommending reduction of fat consumption, also advise avoidance of dietary trans-fats, reducing the intake of saturated fats, and preferring mono or poly-unsaturated long-chain fats [5]. Achieving these goals remains a challenge for physicians and patients. Furthermore, the level of evidence for these recommendations is backed-up by single randomized clinical trials [6] and mostly relies upon large non-randomized observational studies [7], which suffer from confounding [8] and difficulties in quantitatively measuring dietary intake [9].

In addition, other factors linked to dietary consumption which are less likely to be captured in epidemiological studies have emerged as being associated with the risk of atherosclerotic CVD, and include the type of food availability, personal knowledge of the impact of diet on health, and socio-economic status [10]. As an example, the availability of processed foods is associated with an increased risk of CVDs [11]; this is the case of fat-rich processed meats, whose consumption increases the risk of CVD as compared to fat-rich unprocessed meats [12], fatty fish, and poultry [13], whose consumption is not considered a CVD risk modifier.

Furthermore, in modern societies, we are continuously exposed to postprandial lipemia (PPL) [14], a condition that appears to be causally related to the risk of coronary artery disease [15], myocardial infarction, ischemic heart disease, and ischemic stroke [16]. The mechanisms by which an exaggerated PPL links to CVDs include the fostering of endothelial dysfunction [17,18,19], arterial inflammation, and a pro-atherogenic activation of myeloid cells [18]. In addition, the magnitude of PPL in response to high-fat- based meals in humans appears to be significantly affected by the taxonomic composition of intestinal microbiota [20], which also cross-talk with hematopoietic niches [21], ensuring the activity of the innate immune check-points in the intestine and lymphatics. Once absorbed in the intestine, the majority of dietary fats cross over a complex surveillance system (including the cells patrolling at the interface between enterocytes and lacteals like the mesenteric lymph node (MLN), as opposed to the carbohydrates, sugars, dietary amino acids, protein-rich foods and other dietary components that are believed to exert other pathophysiological mechanisms that do not engage these immune checkpoints [22,23,24,25]. For example, an elevated intestinal absorption of sugars promotes systemic inflammation by directly disrupting the intestinal barrier [26]. Observational studies have provided contrasting results, with some studies being in favor of t pro-inflammatory effect of dietary fats and others unable to show any effect. In this review, we aim at critically revising the available evidence and providing a platform to reconcile these findings.

## 2. Dietary Fats, Inflammation and Atherosclerosis

In the human body, dietary fats face a complex metabolic journey involving a number of cellular checkpoints (Figure 1).

Dietary fats (triglycerides, phospholipids and cholesterol) are digested in the upper part of the small intestine by the activity of multiple lipases and then absorbed by the enterocytes [27,28]. In addition, gut resident bacteria can contribute to circulating fats principally by producing short-chain fatty acids (SCFAs) from fibers/complex carbohydrates. For example, *Faecalibacterium prausnitzii* ferments fibers present in the food matrices of fatty foods (e.g., avocados, tree-nuts and peanuts, where over a third of carbohydrates are fibers) [29] and is the major producer of butyrate. Butyrate is known to regulate hematopoietic activity [21] and to control myeloid pro-inflammatory skewing [30], exerting anti-atherogenic properties [31]. Vice versa, *Ruminococcus bromii*, which is reduced in subjects with atherosclerosis [32], metabolizes complex carbohydrates (that are present in low-fat foods, including pinto beans, whole grains, and nuts with considerable proportion of fats) into propionate. This SCFA promotes insulin sensitivity and reduces the atherosclerotic burden in mice [33]. Beside the production of SCFAs, other gut microbial species express enzymatic systems that metabolize dietary lipids into inflammatory molecules. Among them, trimethylamine (TMA) lyase, an enzyme that converts dietary phosphatidycholine and choline into TMA, is peculiarly expressed by *Eggerthella lenta* and *Eggerthella timonensis* [34,35]. Finally, gut resident Gram-negative commensals (e.g., *Escherichia coli, Salmonella minnesota, Salmonella typhimurium*) synthetize lipid-containing molecules, such as lipopolysaccharide (LPS), which promote apoptotic signaling and trigger systemic immune-metabolic derangement and inflammation [36,37,38].

Once within the enterocytes, SCFAs directly reach the portal system and the liver where they are readily metabolized, while the majority of absorbed dietary fats are released into the lymphatic tree in large lipoproteins (chylomicrons and very-low density lipoproteins, VLDL) [39] (Figure 1). Fatty acids deriving from the hydrolysis of dietary triglycerides and phospholipids in the intestinal lumen are chaperoned to the intracellular endoplasmic reticulum (ER); there, diacylglycerol O-acyltransferase 1 (DGAT1) promotes their re-incorporation in triglycerides which are then transferred by the microsomial triglyceride transfer protein (MTTP) to nascent apolipoprotein B. In this way, chylomicrons are released by the enterocytes in their basolateral membrane. A small fraction of absorbed cholesterol is esterified by acyl CoA-transferase (ACAT) and packaged into chylomicrons. In the Golgi, other apolipoproteins, including apoCIII, apoCII, apoAV, and apoAIV, are added to chylomicrons, which then enter the bloodstream via the thoracic duct and will be eventually taken up by liver (Figure 1).

In addition to chylomicrons, the intestine also produces a small fraction of high density lipoproteins (HDL), through the activities of ATP binding cassette transporter A-1 (ABCA1) and phospholipid transfer protein (PLTP), which transfer cholesterol and phospholipids to apolipoprotein A-I [40]. It has been proposed that a fraction of HDL produced by the intestine moves to the liver through the portal system and antagonizes the binding of LPS to toll-like receptor 4 (TLR4) on the membrane of Kuppfer cells, liver-resident macrophages involved in the defense against gut-derived exogenous molecules [41], thus preventing the recruitment of pro-inflammatory myeloid cells [42] (Figure 1). Although further investigations are required, these findings are in line with the known anti-inflammatory function of HDL [43].

To date, the contribution of each single dietary fat in these multiple systems remains unclear. Moreover, a high quantity of data has been produced linking each single dietary fat to key molecular mechanisms and atherosclerosis, but principally providing scattered evidence from experimental models with poor translation to clinics. These aspects will be reviewed below.

### 2.1. Short Chain Fatty Acids

SCFAs (butyrate, acetate, and propionate) originate mostly from the fermentation of fibers and complex carbohydrates (that are abundant in vegetables, fruits, legumes, and whole grains), a process that is triggered by some gut resident bacteria [44,45]. SCFAs participate in multiple processes and their interaction with different receptor systems that are expressed in immune cells [46,47]. Accumulating experimental data have been used to analyze the immune-inflammatory cellular downstream effects of SCFAs using either purchased-SCFAs bound to bound to bovine serum albumin (BSA) or SCFAs from direct fermentation of commensals. In fact, SCFAs, by binding G protein-coupled free fatty acids receptors (FFARs; in particular isoform 2) [48], are significantly expressed in immune cells [49] and promote mechanisms engaged in the resolution of vascular inflammation. By interacting with these receptors, in one study, butyrate-BSA inhibited reactive oxygen species production in neutrophils, decreased the production of inflammatory cytokines (including monocyte chemoattractant protein-1 (MCP-1) and interleukin n-6 (IL-6)) [50]) and down-regulates the pro-inflammatory switching of macrophages [50]. Similarly, in an independent study, butyrate directly produced by gut commensals promotes the anti-inflammatory activity of regulatory T cells (Tregs) [51] (Figure 2).

By interacting with FFAR2, acetate-BSA promotes the phagocytic activity of macrophages, inhibits LPS-induced secretion of inflammatory cytokines by mononuclear cells [52], and triggers the production of oxygen free radicals at the sites of inflammation. Furthermore, mice fed a fiber-enriched diet with acetic acid in water ad libitum were protected against *C. difficile* infection due to interaction of gut commensal-derived acetate with FFAR2 that promotes the switching of the NOD-like receptor protein 3 (NLRP3)-inflammasome [53] (a central pathway in immune-metabolic derangements [54] and atherosclerosis [55]) (Figure 2).

In addition to FFARs, SCFAs also target histone deacetylases (HDACs), a group of deacetylating enzymes that regulate gene expression by removing acetyl groups from both histone and non-histone protein complexes in different genomic regions. Interestingly, butyrate and propionate reduce the production of pro-inflammatory IL-8 in activated endothelial cells by inhibiting HDAC activity [56] (Figure 2). With the same mechanism, butyrate (administered in mice in drinking water ad libitum) down-regulates LPS-induced secretion of pro-inflammatory mediators (such as nitric oxide, IL-6, and IL-12) by intestinal macrophages [57] (Figure 2).

Other in vivo observations in germ-free apoE-/- mice fed a fiber-rich diet and then inoculated with *Roseburia intestinalis* (a butyrate producer) show an increase in intestinal gluconeogenesis, reduction of LPS in the blood and aortic atherosclerotic lesions, decreased expression of inflammatory cytokines, and macrophage accumulation [58,59] (Figure 2). Similarly, apoE-/- mice fed a purified diet low in fibers and supplemented with propionic acid in the drinking water showed moderate cardiac hypertrophy, reduced aortic atherosclerosis and fewer effector lymphocytes (Teff) in the atheroma as compared to apoE-/- mice not receiving propionate supplementation [33]. By contrast to butyrate and propionate, intra-gastric infusion of acetate promotes glucose intolerance [60] and favors the polarization of naïve CD4 + T cells towards Teff, while inhibiting the immune suppressive activity of Treg [61] (Figure 2).

### 2.2. Medium and Long Chain Fatty Acids

Fatty acids can be defined as medium-chain, with 6–12 carbons, and long-chain, with up to 22 carbons. Dietary medium-chain fatty acids are majorly present, in the form of triglycerides, in coconut oil, palm kernel oil, and in dairy products; long-chain fatty acids are more abundant in vegetable oils, tree nuts, peanuts, and fish (principally salmon, tuna, mackerel, and sardines). Fatty acids can be saturated (SFA), monounsaturated (MUFA), or polyunsaturated (PUFA). Among PUFAs, those with a first double bond on the third carbon are referred to as n-3, whereas those with a first double bond on the sixth carbon are called n-6.

Both the degree of saturation and the chain length influence the effect of a specific fatty acid in the immune-inflammatory pathways. In adipocytes, for example, saturated lauric acid, myristic acid, and palmitic acid (12, 14, and 16 carbons, respectively) activate inflammatory genes [62], whereas stearic acid (a SFA with 18 carbons) does not [63]. The scenario, however, is complex, since contrasting data from the literature impede clearly discriminating the pro- or anti-inflammatory properties of medium-chain fatty acids.

SFAs stimulate the inflammatory activation of macrophages by a process that involves TLR4, a pattern recognition receptor that plays a key role in the innate patrolling of bacterial pathogens, including LPS. In fact, the activation of TLR4 by SFA induces an over-activation of IL-6 and TNF-α inflammatory genes through a nuclear factor κB (NFκB)–dependent mechanism [64,65,66]. These effects are reduced by docosahexaenoic acid (DHA, an n-3 PUFA with 22 carbons), which inhibits NFkB and down-regulates two isoforms of the TLR family, TLR2 and TLR6, by impeding their dimerization [67]. Similarly, in vitro, DHA and eicosapentaenoic acid (EPA, an n-3 PUFA with 20 carbons) inhibit the LPS-induced gene expression of cyclooxygenase-2 (COX-2), which is instead increased by treatment with lauric acid [68]. The effect of SFAs on TLR4 signaling is not due to a direct interaction [69], but it requires bacterial LPS binding to CD14, a complex that promotes the endocytosis and lysosomal degradation of TLR4 [70]. Palmitic acid prolongs the activation of TLR4 in macrophages pre-stimulated with LPS, enhancing the production of pro-inflammatory cytokines (MCP-1 and TNF-α) and apoptotic signals that are mediated by the ER [71]. Vice versa, macrophages pre-stimulated with palmitic acid and then treated with palmitoleic acid (MUFA) showed reduced pro-inflammatory activation [72].

The TLR4-mediated engagement of NFkB is tightly linked with the activation of NLRP3 in macrophages [73]. Palmitic acid fosters the activation of NLRP3, by inhibiting adenosine monophosphate-activated protein kinase (AMPK) [71], whereas unsaturated fats inhibit the activation of the inflammasome [74], reverting the apoptotic signals triggered by ER stress in macrophages [75]. In addition, maresin-1 (a DHA-derived metabolite produced by 12-lipoxygenase) and resolvin D2 (produced by 15-lipoxygenase) down-regulate the NLRP3 pathway with subsequent inhibition of caspase-1 and reduction of IL-1β secretion [76] (Figure 2). By this mechanism, maresin-1 and resolvin D2 induce an anti-inflammatory phenotype in macrophages from apoE-/- mice, an effect that results in the stabilization of the atherosclerotic lesion [77]. As compared to the n-3 series, n-6 PUFAs clearly demonstrated pro-inflammatory effects. Arachidonic acid (ARA) (a 20 carbon, n6 PUFA that can be found only in animal-derived foods) promotes oxidative metabolism [78] and stimulates the release of IL-6 and TNF-α by macrophages through the production of two downstream products of ARA, the leukotriene B4 (LTB4) and the prostaglandin E2 (PGE_2_ [79]). Also, further in vitro experiments postulated that ARA is able to activate the Janus–Kinase pathway in macrophages, inducing cell cycle arrest [80]. In contrast to this pro-inflammatory vision, ARA might also support the efficiency of neutrophils in sites of inflammation [81] (Figure 2). In fact neutrophils, once activated at the site of inflammation through the granulocyte-colony stimulating factor (GM-CSF), increase the uptake of the ARA contained in triglycerides by fatty acid transport protein 2 (FATP2) and convert ARA to the PGE_2_ that promotes the suppression of CD8^+^ cytotoxic T cells [82].

All these cellular pathways have been described in vitro, but the understanding of the mechanisms by which dietary fats are absorbed might allow us to better define how they can exert such effects in vivo. The intestinal absorption of medium- and long-chain fatty acids depends on the length of their aliphatic tails. After hydrolysis of their triglyceride precursors, medium-chain caprylic acid (8 carbons) and capric acid (10 carbons) are absorbed by enterocytes and bind to albumin, being then directly transported to the hepatic portal system [83] (Figure 1). Vice versa, long-chain fatty acids are packaged into chylomicrons in the enterocytes, secreted into the lacteals, enter the intestinal villi, and pass through the MLN before reaching circulation through the thoracic duct (Figure 1). Within this system, long-chain fatty acids in chylomicrons undergo the surveillance of C-X3-C motif chemokine receptor 1 (CX3CR1)-expressing macrophages residing in the villi, the CD103/CD11b expressing DCs (representing up to 80% of the total population of gut resident DCs) [84,85], and the innate lymphoid cell family member 2 (ILC2) [86] (a subset that uses dietary fatty acids as fuel for cellular fatty acid oxidation and energy production to produce IL-5, IL-9, IL-13 against *Trichuris muris* helminth infection [87]). (Figure 3).

Chylomicrons engage the endothelial cells of lymphatic vessels to produce chemokines involved in the activation of effector T-helper 17 cells [88,89]; then, once in the MLN, chylomicrons induce macrophages to secrete pro-inflammatory cytokines and to switch into pro-inflammatory foam cells [90] (Figure 3). Furthermore, the inflammatory cascade involving the metabolic conversion of PUFAs to arachidonic acid (AA) produces different mediators promoting the activity of ILC2 [91,92,93]. Notwithstanding whether these effects are actually mediated by (and which type of) dietary fatty acids remains a matter of debate today. Mineral oils show pro-inflammatory effects in peritoneal macrophages, triggering caspase-1 and NLRP3 activation, whereas some vegetal oils induce foam cell formation and cell death via caspase-3 cleavage-dependent mechanisms [94]. However, macrophages not only take up fatty acids deriving from the hydrolysis of triglycerides contained in chylomicrons and VLDL [95], thus inducing pro-atherogenic effects in endothelial cells [17] and monocytes [96], but LPS produced by gut commensals [97]. It is therefore plausible that macrophages might be activated by this bacteria-derived component, as suggested by the observation that gut-derived LPS and TLR4 co-localize with CD68, a marker of macrophages, in carotid atheromas [38].

### 2.3. The Contribution of Dietary Cholesterol as Compared to Serum Cholesterol

The intestine is the source of exogenous cholesterol, representing up to 30% of the total cholesterol pool in the body [98,99]. Cholesterol in the intestine principally derives from the enterohepatic circulation (bile) and, secondly, from dietary sources (principally from animal and dairy food products). In the intestinal lumen, esterified cholesterol is hydrolyzed by the pancreatic cholesteryl ester hydrolase, producing free cholesterol [98]. Free cholesterol is then emulsified, along with other lipids and vitamins, into micelles and absorbed by the enterocytes through multiple transport systems such as Niemann-Pick C1-like 1 (NPC1L1) andABCG5/ABCG8. After absorption, free cholesterol is re-esterified by ACAT and packaged into chylomicrons [98]. In the circulation, chylomicrons undergo the activity of lipases, which hydrolyze their lipids and reduce their diameter, generating chylomicron remnants which are removed from the circulation by the liver [98]. There, cholesterol is principally rewired to the intestine via the enterohepatic circulation, while a modest fraction is secreted in the systemic circulation, packaged into VLDL. VLDL are significantly produced during PPL and, following the activity of lipases, they become low density lipoproteins (LDL), smaller in diameter, and with a predominant cholesterol content. Despite the average content of Western-style meals being estimated between 20 and 40 g of total fats/meal and three-four meals/day being typically consumed [100,101], given the actual content of cholesterol in the majority of foods (by 4 to 700 mg per quantity of food consumed [102,103]), it appears clear that the dietary source of cholesterol is minor as compared to that re-cycled through this complex metabolic system. Historical data actually demonstrated that there is a poor effect of the physiological consumption of cholesterol by diet and changes in serum cholesterol [104]. In line with this, both the American Heart Association Guidelines of 2013 [105] and the more recent indications of the European Society of Cardiology (ESC) and European Atherosclerosis Society (EAS [5]) toned down the magnitude of the effect and the level of evidence of reducing dietary cholesterol intake for CVD prevention.

Notwithstanding that, by virtue of their diameter and because they are significantly smaller than chylomicrons, VLDLs (30–70 nm) can enter the intima, and it is plausible that dietary cholesterol accumulating over time, can exert pro-inflammatory effects in the vasculature (Figure 2). When excessively produced during PPL, VLDL can foster atherogenic effects in endothelial cells by enhancing the expression of chemokine receptors and inducing apoptotic signals [17]. PPL extends the effects mediated by VLDL and furthermore promotes the accumulation of cholesterol in LDL. LDL are smaller than VLDL (20–30 nm in diameter) and their atherogenic potential is even higher [106]. Cholesterol can be oxidized into different types of oxysterols by a number of cardiovascular risk determinants as well as by factors related to the industrial processing of foods [3,11]. Oxysterols contribute to the formation of modified LDL (namely oxidized LDL, oxLDL), which are taken up by macrophages in the atheroma. Within cells, the crystallization of excess cholesterol occurs, thus further increasing its atherogenic potential and the ability to evoke the inflammatory activation of Teff [107,108,109] and the induction of NLRP3 [110,111] (Figure 2). Acute exposure of macrophages to oxLDL prolongs these mechanisms by inducing epigenetic priming of a complex set of inflammatory players [112]. Also, NLRP3 undergoes this epigenetic long-lasting activation, a process that has been described to favor an inflammatory phenotype of the myeloid hematopoietic immune compartment [113].

A small amount of robust experimental evidence suggests the intra-cellular effect of cholesterol in promoting the commitment of hematopoietic stem cells towards the expansion and pro-inflammatory activation of the myeloid compartment [114]. In fact, the apoE-/- mouse model of hypercholesterolemia actually displays aberrant hematopoietic commitment, a phenomenon that is mainly attributed to the remodeling of membrane lipid rafts and the over-expression of the GM-CSF receptors CD131 and CXCR4 (physiological regulators of HSC egress from medullary niches) [115] (Figure 2).

## 3. Data Linking Intake of Dietary Fats, Markers of Inflammation, and Risk of CVD

The evidence about the molecular and cellular mechanisms by which dietary fats participate in atherogenesis built up the rationale to unveil the connections between different dietary fats, the markers of systemic inflammation, and the risk of CVDs. Despite this aspect having been extensively discussed, data from both epidemiological studies and interventional clinical trials are, however, scarce and heterogeneous. Here, we will provide a separate discussion for results from epidemiological studies and interventional clinical trials.

### 3.1. Epidemiological Studies

An overall summary of the results from epidemiological studies on the relationship between the pro- or anti-inflammatory effects of dietary fats and CVD risk is reported in Table 1.

The association between the intake of SFAs, markers of inflammation, and CVD has been assessed in recent epidemiological studies. In two independent cohorts (the Nurses’ Health Study and the Health Professionals Follow-up Study (HPFS)), increasing dietary intake of saturated and trans-fats was significantly associated with a higher risk of CVDs over a long-term follow-up (1980–2012), with dietary assessment evaluated every four years [119]. Another analysis from these two studies reported that a higher consumption of saturated dietary fats was not associated with higher circulating levels of IL-6 and C-reactive protein (CRP) both in women and men with a history of CHD, whereas this association was observed in subjects without previous CHD [143]. A lack of association was also found among older adults at elevated CV risk from the Health ABC Study [138], where the relation between elevated risk of heart failure and CHD and higher levels of inflammatory markers was independent from the intake of dietary fats [144,145]. In the Scottish Lothian Birth Cohort 1936 study, a higher consumption of saturated fats correlated with elevated CRP (but not with higher fibrinogen or IL-6) in older subjects; however, this association was significant in subjects with hypercholesterolemia, but not in subjects with previous CVDs and ischemic stroke [146]. In another study, in men with acceleration of aortic pulse wave velocity (aPWV) no relation was found between the dietary intake of saturated fats and CRP plasma levels [120]. In the SUN cohort, a higher intake of saturated fats, which contributed to higher inflammatory potential of diet (expressed by the dietary inflammatory index or DII) predicted higher occurrence of CVDs over an eight-year follow-up [147]. It has to be acknowledged that the lack of information about quantitative changes in circulating markers during follow-up does not allow to link the intake of dietary fats with systemic inflammation. The large Prospective Urban Rural Epidemiology (PURE) study, assessing the predictive role of dietary intake of fats on the occurrence of CV events over eight-year follow-up [148], found that lower intake of SFAs was associated with lower risk of ischemic stroke, whereas it did not associate with the risk of myocardial infarction. Also in this study, the lack of information on inflammatory markers does not allow to draw any conclusions.

The increased consumption of total trans-fats has been historically associated with an increased risk of CHD [119,121]. However, recent evidence attributed opposing effects to the two main classes of trans-fats. In fact, higher plasma phospholipids content of ruminant-trans fatty acids (rTFA), which are naturally found in dairy foods and meats, was associated with reduced CVD risk factors and with higher adiponectin levels (an anti-inflammatory adipokine). Conversely, a higher content of industrial-trans fatty acid (iTFA) (such as trans elaidic fatty acid) in plasma phospholipids was associated with higher CVD risk factors [116], although conflicting results have been reported. In the Akershus Cardiac Examination 1950 study, reduced intake of iTFA was associated with increased plasma levels of CV risk factors (triglycerides, fasting glucose, blood pressure and CRP) [117]. These discordant observations might be in part explained by the matrix effect, as multiple components, preservatives, and industrial processes in food might affect the link between inflammation and CVD as compared to a single dietary fatty acid [11]. This hypothesis might be also applied to previously discussed SFAs from non-processed dairy products whose consumption, despite not being associated with CVD risk [11], correlates with the reduction in inflammatory marker levels, such as CRP, IL-6, and TNF-α [149].

Several epidemiological studies have provided discordant data also for the association between dietary intake of unsaturated fats and CVD. The Nurse Health Study showed that a higher intake of both MUFAs and PUFAs was associated with reduced CV mortality; the Health Professional Follow-up study confirmed this association only for PUFAs [119]. These two large studies did not evaluate whether these results might be correlated to significant changes in markers of low-grade inflammation. In the Caerphilly Prospective study, the increased dietary intake of PUFAs was associated with reduced CRP and fibrinogen levels and with a higher aPWV. Vice versa, the intake of MUFAs, which are associated with fibrinogen levels, did not predict increase in aPWV [120]. The observation regarding the intake of PUFAs was also confirmed in the Whitehall study of London civil servants [150], while no information for the intake of MUFAs is available in this study.

The conflicting results reported for MUFAs might be attributed to the type of food source, as only MUFAs from animal sources (such as beef, pork, and processed meat, which also contributes to saturated fats intake), but not those from plant sources are associated to 16% higher risk of CVD mortality [128]. In addition, consumption of olive oil and nuts, which are the main plant food sources of MUFAs, is associated with both lower occurrence of nonfatal CVD events and lower inflammatory markers (IL-6) levels [126,127].

Among PUFAs, whether the potential anti-inflammatory effect is to be attributed to the n-3 or n-6 series is an actual matter of study. The adherence to the Mediterranean diet-based model of the PREDIMED study promoting the consumption of n-3 enriched based foods, was associated with reduced DII [130]. This finding was also reported in the PRE-diabetes and type 2 DIAbetes (SPREDIA-2) study, in which a higher consumption of n-3 PUFAs associated with reduced CRP levels [151]. In contrast, in patients in secondary prevention from the Western Norway B Vitamin Intervention Trial, a higher consumption of n-3 from either fatty fish or fish oil did not protect from recurring coronary events, nor it was associated with a significant reduction in CRP levels [136]. Vice versa, in men in primary prevention from the Health Professionals Follow-up Study HPFS, n-3 PUFAs from both seafood and plant sources predicted lower incidence of coronary events independently of the concomitant dietary intake of n-6 fatty acids, despite the effect on inflammatory markers not being investigated [137]. Also, in healthy women of the Nurses’ Health Study I cohort, the intake of α-linolenic acid was inversely related to plasma concentrations of CRP and E-selectin, and EPA and DHA were inversely related to other markers of vascular endothelial dysfunction (ICAM-1 and VCAM-1) [132]. In the HPFS and the Nurses’ Health Study II, a higher intake of EPA and DHA was associated with reduced plasma levels of both soluble isoforms of the TNF receptor, despite non-significant trends towards the reduction of CRP levels [133]. Of note, these associations were significant even in subjects with an elevated dietary intake of the n-6 series, suggesting that these fatty acids might not contrast the plausible anti-inflammatory effect of the n-3 series [133]. By contrast, in over three thousand community-dwelling Japanese individuals from the Hisayama Study, the decrement in the EPA/AA ratio significantly increased both the risk of occurring CVD and the serum levels of CRP [134]. In addition, low fish consumption, low EPA and DHA intake and a low intake of AA were also associated with higher levels of inflammatory markers in subjects with coronary artery disease [135]. Nuts represent an alternative food source of PUFAs, and accumulating evidence supports their anti-inflammatory and cardioprotective effects. High nut consumption (>120 g/week), particularly MUFA- and PUFA-enriched tree nuts and ground nuts, is associated with reduced CV risk factors, improved postprandial lipemia and significant reduction in CVD mortality [131]. Furthermore, the reduced risk of CVD mortality attributable to nut consumption (>2 times/week), was linked, although in a modest proportion (17.8%), to lower hs-CRP levels [129].

Dietary cholesterol present in multiple processed and non-processed food patterns, including eggs, butter, beef, cheese and shrimp, deserves a separate discussion. The established pro-inflammatory effects of cholesterol have been hardly confirmed in epidemiological studies, which report contrasting data on the relationships between dietary consumption of cholesterol, markers of inflammation and CVD risk [152]. In fact, the association between dietary cholesterol from eggs and markers of systemic low-grade inflammation is not clear, with studies in both healthy subjects and patients with type 2 diabetes reporting contrasting findings [139,140]. Bechthold et al. showed no correlation between the highest (75 g) and lowest intake of eggs (0 g) and the risk of CHD or stroke. In a dose-response sub-analysis, increased increments of egg intake (50 g) were predictive of higher risk of heart failure, but not CHD or stroke [153]. This observation was replicated in the Kuopio Ischaemic Heart Disease Risk Factors Study, where cholesterol intake from eggs was not associated with the risk of ischemic stroke [141]. Similarly, in the Nurses’ Health Study and in the HPFS, the intake of dietary cholesterol (consumed as one egg daily) was not associated with increased risk of coronary artery disease in healthy men and women [142]. These data might be principally explained by the limited contribution of dietary cholesterol as compared to the endogenous fraction, which has been for long time the principal target of all the most effective pharmacological options. Furthermore, it cannot be ruled out that again the food matrix effect might reconcile the inflammatory effect of specific atherogenic components of eggs. This is the case of the immune-inflammatory and atherogenic trimethylamine N-oxide (TMAO) [32,154], which is significantly absorbed following the consumption of a fixed quantity of eggs, and its plasmatic concentration seems to depend on the abundance and activity of specific gut resident bacteria [155].

### 3.2. Interventional Clinical Trials

Table 2 reports a summary of the results from clinical trials, testing the impact of dietary fat consumption on markers of inflammation and CVD risk factors.

Lower levels of adhesion molecules and CRP were observed when 8% of energy from trans-fats was replaced with the same amount of energy from MUFAs [156]. Vice versa, the substitution of SFAs (stearic acid) with trans-fats did not exert effects on inflammatory marker levels (CRP and IL-6) [157]. This study, however, did not discuss whether substitution with either rTFA or iTFA would have exerted different effects.

Also, replacing 8% of energy from digestible carbohydrates with the same amount of energy from either medium-chain or long-chain SFAs (stearic acid) did not reduce CRP and IL-6 levels over a five-week period [156,162]. In an independent cross-over randomized trial, consumption of SFAs from butter or cheese did not increase hs-CRP levels as compared to an isocaloric carbohydrate-enriched dietary regimen, despite inducing an increase in LDL-C [161]. A meta-analysis of 16 trials assessing the effects of the consumption of coconut oil (containing elevated amount of SFAs) or other fats found that coconut oil increased cholesterol, but had no effects on markers of systemic inflammation [181]. Similarly, CRP did not increase in cross-over-based trials comparing a high-cholesterol high-fat diet with a low-cholesterol high-fat diet, both in healthy subjects [178] and in patients with type 2 diabetes [179].

Trials comparing the substitution of SFAs with MUFAs reported a beneficial effect on LDL-C levels, but no changes in CRP and other inflammatory markers (including IL-6) [156,162,163]; other trials could not confirm this effect on LDL-C levels [164,182]. An increased intake of PUFAs was associated with lower risk of coronary artery disease [166], despite the potential pro- or anti-inflammatory effect not being assessed.

Both LDL-C-lowering effects and a trend towards CRP reduction were observed in studies evaluating consumption of n-6 PUFAs in place of SFAs [161,163]. Similarly, the replacement of SFAs with n-3 or n-6 PUFAs exerted significant reductions in CRP levels and lowered total cholesterol, LDL-C and triglycerides [160]; of note, these reductions were more robust following the substitution of SFAs with n-3 rather than n-6 PUFAs. In dyslipidaemic patients, n-6 PUFA supplementation lowered total cholesterol concentration from baseline without any effects on inflammatory markers [167].

N-3 supplementation trials failed to provide conclusive evidence on their effects on inflammation markers and CVD risk factors [172]. For example, in dyslipidaemic patients, the significant reduction in CRP and IL-6 observed with n-3 supplementation was not paralleled by a robust improvement in plasma lipids [167]. The REDUCE-IT trial demonstrated that the intervention with 4 g/day of a particular formulation of EPA provided a robust reduction in CRP levels and a significant reduction in the risk of atherosclerotic cardiovascular events in patients at elevated CV risk despite receiving the maximally tolerated statin therapy [168]. Besides these pharmacological findings, either nutraceutical or dietary supplementation with unsaturated fats and n-3 series exerted contrasting results on markers of inflammation and CVD risk. Recent data from the VITAL Research Group indicate that supplementation with 1 g/day of marine n-3 fats did not reduce the incidence of major cardiovascular events. A modest reduction in the risk of myocardial infarction was reported in subjects of the same study reporting dietary fish consumption less than 1.5 servings/week [183], but the lack of data on markers of inflammation does not allow the researchers to conclude whether this finding might be proportional to from the results of REDUCE–IT.

In parallel to these data, a beneficial effect of the dietary consumption of the n-3 series on markers of cellular inflammation has been proposed For example, increased n-3 consumption from nuts lowered blood pressure, plasma cholesterol levels, reduced markers of DNA oxidative damage in peripheral leukocytes, and improved endothelial function in subjects with metabolic syndrome [169,170,171,173]. Vice versa, in obese subjects, dietary n-3 supplementation at different dosages (although lower than that used in REDUCE-IT) had opposing effects on inflammatory markers and CV risk factors. At lower doses, when consumed as supplemented milk, there was no effect on inflammation, but a significant increase in HDL-C was observed [174]. At higher doses, obtained with n-3 rich flaxseed flour, reductions in CRP and serum amyloid A were reported in obese patients without any effects on body weight and other cardio-metabolic markers [165]. Furthermore, n-3 PUFA supplementation favorably influenced arterial stiffness and hs-CRP compared to corn oil supplementation in small sized trials, although not to a statistically significant degree [176]. Similarly, CRP was significantly reduced in statin-naïve subjects with dyslipidemia receiving n-3 fatty acids [176]. The consumption of n-3 PUFA-enriched foods (fish oil or fruit juice or fish pate), providing approximately 1 g EPA + DHA, reduced interferon-γ (IFN-γ) levels without any change in lipid profile [175].

The effects of n-3 PUFA are also influenced by the type of PUFA supplemented: a head-to-head comparison of the effects of EPA and DHA showed higher beneficial effects on inflammatory marker levels and CVD risk factors in the DHA-treated group, even though an increase in LDL-C was observed [184].

## 4. Conclusions

Despite the strong evidence at the cellular and molecular level, the relationship between dietary fats, immune-inflammation and atherosclerotic CVD is not well established at the clinical level [185]. Methodological and technical difficulties are, however, to be acknowledged. In fact, interventional trials, although designed with the attempt to provide strong causal inference, are oftentimes sample-sized, have short follow-up, have high attrition rates and have to cope with the difficulty of the standardization of the measurement of biomarkers. In parallel, observational studies are overly of misinterpreted dietary intakes, and the lack of sufficient follow-up oftentimes affect the robustness of the outcomes. Despite these difficulties, the principal take-home message from this complex background is that the paradigms the evolved during the last decades addressing pro-inflammatory effect to dietary SFAs, cholesterol and trans-fats while an anti–inflammatory potential to dietary MUFAs, PUFAs or n-3 probably need to be re-challenged with respect to the international recommendations for CVD prevention. As a matter of fact, recent large, multi-center observations questioned this archetype [10,186], supporting that additional factors, related to the individual environmental exposure, can be more likely effective. Among them, the matrix effect (the sum of the content in micro-, macronutrients and additives in food patterns) principally determines different associations between dietary fats and CVD [10,11]. Also, the metabolic postprandial response to foods, in particular to highly-caloric dietary fats, was recently demonstrated to vary largely among individuals because of strong interference from the gut [187,188]. Finally, even more recently, individual signatures of gut microbiota were connected to systemic markers of immune-inflammation in determining individual profiles of postprandial response to foods [20]. Altogether, these important observations suggest that the connection between dietary fats and CVD should be studied taking into account both the immune-inflammatory potential of these dietary sources and the individual predisposition for their metabolism. Under this perspective, a more personalized approach to diet could be pursued.

## Figures and Tables

**Figure 1 nutrients-13-03768-f001:**
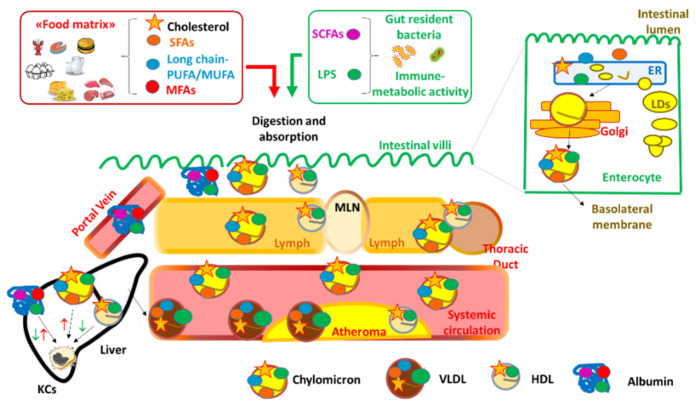
The routes for the absorption and the principal immune-inflammatory engagements of dietary lipids in intestinal villi, in the lacteals, MLN up to the liver. MLN = mesenteric lymph node; LDs: lipid droplets; SFAs: saturated fats; MUFA: mono-unsaturated fats; PUFA: poly-unsaturated fats; SCFAs: short chain fatty acids; MFA = medium chain fatty acids; LPS: lipopolysaccharide; ER: endoplasmic reticulum; LDs = lipid droplets; CM: chylomicrons; VLDL = very low density lipoproteins; HDL = high density lipoproteins; DC = dendritic cells; KCs = Kuppfer cells. Upward red arrows indicate activation of a cell or a pathway; downward green arrows indicate inhibition or regulation of a cell or a pathway. Both upward red and downward green arrows for SCFAs and MFAs carried by albumin in the liver indicate contrasting evidence depending on the type of dietary fat.

**Figure 2 nutrients-13-03768-f002:**
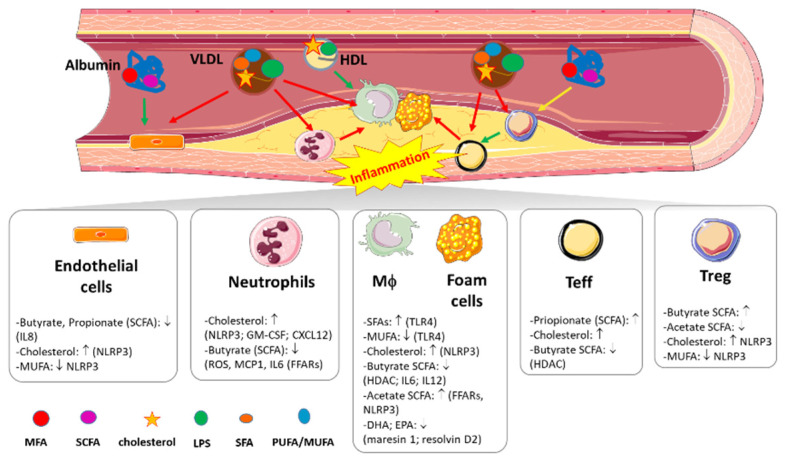
Principal immune-inflammatory pathways elicited by dietary fats in atherosclerosis. VLDL = very low density lipoproteins; HDL = high density lipoproteins; LPS = lipopolysaccharide; Mϕ = macrophages; Teff = effector T cells; Treg = regulatory T cells; SFAs = saturated fatty acids; SCFAs = short chain fatty acids; MUFA = mono-unsaturated fatty acids; PUFA = poly-unsaturated fatty acids; DHA = docosahexaenoic acid; EPA = eicosapentaenoic acid; NLRP3 = NOD-like receptor protein 3; GM-SCF = granulocyte-coly stimulating factor; CXCL12 = C-X-C motif chemokine ligand 12; ROS = reactive oxygen species; MCP-1 = monocyte chemoattractant protein-1; FFARs = free fatty acids receptors; IL- = interleukin; HDAC = histone deacetylases. ↑: indicates an activated cell or pathway; ↓: indicates an inhibited cell or pathway.

**Figure 3 nutrients-13-03768-f003:**
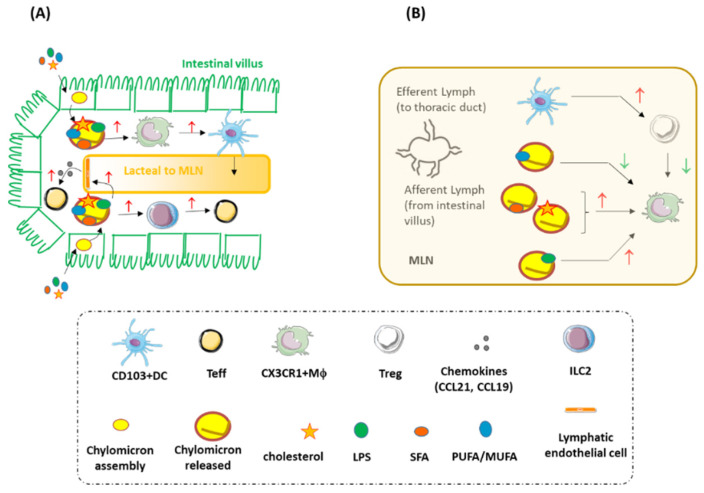
Summary of the immune-inflammatory pathways targeted by dietary fats in the intestinal villus (**A**) and in the MLN (**B**). LPS = lipopolysaccharide; CX3CR1 = C-X3-C motif chemokine receptor 1; Mϕ = macrophages; Teff = effector T cells; Treg = regulatory T cells; CCL- = C-C motif ligand-; ILC2 = innate lymphoid cell sub-type 2; SFA = saturated fatty acids; MUFA = mono-unsaturated fatty acids; PUFA = poly-unsaturated fatty acids; ↑: indicates an activated cell or pathway; ↓: indicates an inhibited cell or pathway.

**Table 1 nutrients-13-03768-t001:** Summary of the data from epidemiological studies on the association between dietary intake of fats, circulating markers of systemic inflammation and risk of CVD. ↑: Indicates data showing a positive association between the dietary fat intake and the outcome (either inflammatory markers or CVD risk factors); ↓ indicates data showing a positive association between the dietary fat intake and the outcome (either inflammatory markers or CVD risk factors). ↔ indicates that there are missing or contrasting data regarding association between the dietary fat intake and the outcome (either inflammatory markers or CVD risk factors).

EPIDEMIOLOGICAL STUDIES		
Dietary fats	Prevalent Effects on Inflammatory Markers	Effects on CVD Risk/Risk Factors
Trans fats	↑ [116] [117] [118] ↓ ↔ [119,120] [121] [122]	↑ [116,119,120,121] [117] [118,123] ↓ ↔ [122]
Saturated fats	↔ [119,120] [121] [124,125]	↑ [119] ↔ [120] [121] [123,124]
Monounsaturated fats	↓ [119] [126,127]	↓ [119] [126,127] ↔ [121] [128] [123]
Polyunsaturated fats	↔ [119] [121] [126] [129]	↓ [119] [121] [123] [126] [130,131] [129]
n-3 and derivates	↓ [129] [132,133,134,135] ↔ [136]	↓ [132,137] [134] ↔ [136]
n-6 and derivates	↔ [119]	↓ [119]
Cholesterol	↑ [125] ↔ [138,139,140,141]	↑ [138] ↔ [139,140,141,142]

**Table 2 nutrients-13-03768-t002:** Summary of data from interventional studies about the association between dietary intake of lipids, circulating markers of systemic inflammation and risk of cardiovascular diseases. ↑: Indicates data showing a positive association between the dietary intervention and the outcome (either inflammatory markers or CVD risk factors); ↓ indicates data showing a positive association between the dietary intervention and the outcome (either inflammatory markers or CVD risk factors). ↔ indicates that there are missing or contrasting data regarding association between the dietary intervention and the outcome (either inflammatory markers or CVD risk factors).

INTERVENTIONAL TRIALS		
Dietary Lipids	Prevalent Effects on Inflammatory Markers	Effects on CVD Risk/Risk Factors
Trans fats	↑ [156] ↔ [157]	↑ [155] [156]
Saturated fats	↑ [158] [159] [160] ↔ [156] [161,162] [163]	↑ [156] [158,161] [159,160,163] ↔ [162]
Monounsaturated fats	↓ [156] [158] ↔ [161,162] [163] [164] [165]	↓ [156] [158,161,162] [163] [164]
Polyunsaturated fats	↓ [159] ↔ [161] [163]	↓ [161] [159,163] [166]
Ω-3 and derivates	↓ [167,168,169,170,171] [165] ↔ [160] [172] [173,174] [175,176]	↓ [160] [168,169,170,171,173] ↔ [167,172] [165,174,175,176]
Ω-6 and derivates	↔ [160] [167]	↔ [160] [167]
Cholesterol	↓ [177] ↑ [178] ↔ [179]	↓ [180] ↑ [179]

## Data Availability

The manuscript does not show data or analyses of data.

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
