# Peer review of "Molecular Immune-Inflammatory Connections between Dietary Fats and Atherosclerotic Cardiovascular Disease: Which Translation into Clinics?"

_nutrients, 2021, doi:10.3390/nu13113768_

Round 1

Reviewer 1 Report

Review. October 5th, 2021.

Paper submitted: “Molecular immune-inflammatory connections between dietary fats and atherosclerotic cardiovascular disease: which translation into clinics?”

Special Issue: Nutraceutical Approaches to Cardiovascular and Metabolic Diseases: Evidence and Opportunities, Journal Nutrients.

General comment

The present article describes nutritional and metabolic aspects of dietary fatty acids and explain the roles of bioactive members of the different fatty acid families in inflammatory processes. The focus is on low grade inflammation related to fat intake and cardiovascular diseases, and the authors aimed at clarifying the whole picture by discussing available literature from cellular and molecular evidence to human epidemiological and intervention data.  

The first part of the paper introduces the epidemiological context and the contribution of dietary strategies to prevent chronic conditions, in particular cardiovascular diseases. In a second part, the authors resume the specific role of different dietary fatty acids (short, medium, long chain fatty acids, cholesterol) in the regulation of inflammatory pathways. And finally, they summarize the results from both epidemiological and interventional studies that investigated either the association between dietary fats intake, the markers of inflammation and CVD risk (epidemiological data), or the causal relationship of dietary fat consumption on markers of inflammation (clinical trials).

The authors concluded that despite strong evidence at the cellular level, the relation between dietary fats, inflammation and atherosclerotic CVD is not yet well established at a clinical level. Additional factors, such as the individual environmental exposure, the food source/ matrix effect and the role of gut microbiota need to be considered when investigating the preventive effect of dietary fats on CVD risk in human.

The submitted paper is clear, well written and well referenced. Such paper is timely and helpful as it deals with a topic where literature has been very inconsistent, while the substantial role of low-grade inflammation in the early-stage pathophysiology of atherothrombotic events has been recognized for decades. The authors bring up to light emerging concept and key elements to be considered, giving us a relevant and holistic approach of the topic.

To conclude, I would only suggest minor changes that may improve the manuscript.

Specific comments on the manuscript

Part 2.1 Short chain fatty acids

This part describes the role of short chain fatty acids (SCFA) in various mechanisms and inflammatory processes. The authors may want to specify and clarify the type of studies they are discussing. For example, some data result from cell-culture experiments, using purchased-free fatty acids BSA, while others reported the effect of SCFA produced by commensal microorganisms during starch fermentation in mice or other animal models. While the goal is to improve the comprehension of current evidence and its translation into clinics, those study details may be of relevance.

 Part 2.2 Medium and long chain fatty acids

Authors reported: “In general, however, it is widely accepted that SFAs have pro-inflammatory effects, while MUFA’s and PUFA’s are anti-inflammatory”. I would suggest toning down that statement. Despite some evidence for pro-inflammatory effects of SFA, literature is not always consistent. Also, among PUFA’s, some members of n-6 family have been clearly identified as pro-inflammatory mediators (such as LA, ARA and eicosanoids). Thus, this paragraph seems to suffer from a lack of data on the effect of PUFA n-6 while the role of (various) SFA as well as DHA and EPA, are well described.

Part 2.3 Dietary cholesterol

This part deals with the atherogenic potential of lipoprotein particle (VLDL and oxLDL). VLDL are triglyceride-rich lipoprotein, and LDL (and oxLDL) carry endogenous cholesterol rather than dietary cholesterol. Although the authors define both exogenous and endogenous pathways, I would suggest modifying the title of this 2.3 part to avoid misunderstanding and confusion. Numerous data recently reported that there is no appreciable relationship between consumption of dietary cholesterol and serum cholesterol.  And in accordance with sciences-based evidence, the American Heart Association eliminated specific dietary cholesterol target in updated guidelines (2013).

Part 3.  

Some statement may need clarification or little changes.

  • Page 9. The authors report “Among the series of SFAs, the increased consumption of total trans-fats has been historically associated with an increased risk of CHD…” : The link between SFAs and trans-fat here is not clear, as trans-fats are part of the unsaturated fatty acid family.
  • Page 9 [… In fact, a higher content of ruminant-trans fat (rTFA), which are naturally found in foods, in plasma phospholipids was associated with reduced CVD risk factors …]. I would suggest being more accurate, as only very few foods are source of such ruminant TFAs: [In fact, a higher content of ruminant-trans fats (rTFA), which are naturally found in dairy foods and meat, in plasma phospholipids was associated with reduced CVD risk factors…]
  • Page 9 […Another example of this hypothesis is that the increased intake of SFAs from non-processed dairy products, despite not associated with CVD risk, correlates with…]. I would suggest modifying the sentence as follow: [… Another example of this hypothesis is that the increased intake of SFAs from whole fat dairy products, despite not associated with CVD risk, correlates with…].
  • Page 10. [ Dietary cholesterol presents in multiple processed and non-processed food patterns, including eggs, butter, beef, cheese butter and shrimps, deserves a separate discussion]. “Cheese butter” is an unknown product among dairy foods. This sentence should be: [ Dietary cholesterol presents in multiple processed and non-processed food patterns, including eggs, butter, beef, cheese and shrimps, deserves a separate discussion].
  • Page 11. Table 2 (interventional studies). It is not clear how the authors resume the contrasting results from literature. For example, the effect of cholesterol is resumed as: “pro-inflammatory” and “increase CVD risk markers”, while both studies [167 and 168] from Blesso et al. showed that an increase in dietary cholesterol resulted in reduction of inflammatory markers. And while the 2 other studies referenced [169 and 170] reported that CRP level did not increase in patients with high cholesterol high fat diet compared to low cholesterol low fat diet (although affecting HDL composition and functionality).

Author Response

REVIEWER#1

Dear Reviewer,

We are grateful for your positive evaluation of our work, we have revised the manuscript according to your suggestions and, please, find below the responses to your comments (in red).

Review. October 5th, 2021.

Paper submitted: “Molecular immune-inflammatory connections between dietary fats and atherosclerotic cardiovascular disease: which translation into clinics?”

Special Issue: Nutraceutical Approaches to Cardiovascular and Metabolic Diseases: Evidence and Opportunities, Journal Nutrients.

General comment

The present article describes nutritional and metabolic aspects of dietary fatty acids and explain the roles of bioactive members of the different fatty acid families in inflammatory processes. The focus is on low grade inflammation related to fat intake and cardiovascular diseases, and the authors aimed at clarifying the whole picture by discussing available literature from cellular and molecular evidence to human epidemiological and intervention data. 

The first part of the paper introduces the epidemiological context and the contribution of dietary strategies to prevent chronic conditions, in particular cardiovascular diseases. In a second part, the authors resume the specific role of different dietary fatty acids (short, medium, long chain fatty acids, cholesterol) in the regulation of inflammatory pathways. And finally, they summarize the results from both epidemiological and interventional studies that investigated either the association between dietary fats intake, the markers of inflammation and CVD risk (epidemiological data), or the causal relationship of dietary fat consumption on markers of inflammation (clinical trials).

The authors concluded that despite strong evidence at the cellular level, the relation between dietary fats, inflammation and atherosclerotic CVD is not yet well established at a clinical level. Additional factors, such as the individual environmental exposure, the food source/ matrix effect and the role of gut microbiota need to be considered when investigating the preventive effect of dietary fats on CVD risk in human.

The submitted paper is clear, well written and well referenced. Such paper is timely and helpful as it deals with a topic where literature has been very inconsistent, while the substantial role of low-grade inflammation in the early-stage pathophysiology of atherothrombotic events has been recognized for decades. The authors bring up to light emerging concept and key elements to be considered, giving us a relevant and holistic approach of the topic.

To conclude, I would only suggest minor changes that may improve the manuscript.

Specific comments on the manuscript

Part 2.1 Short chain fatty acids

This part describes the role of short chain fatty acids (SCFA) in various mechanisms and inflammatory processes. The authors may want to specify and clarify the type of studies they are discussing. For example, some data result from cell-culture experiments, using purchased-free fatty acids BSA, while others reported the effect of SCFA produced by commensal microorganisms during starch fermentation in mice or other animal models. While the goal is to improve the comprehension of current evidence and its translation into clinics, those study details may be of relevance.

We thank the reviewer for raising this important point. We have revised the text as follows (lines 153-158, Page 4):

“...SCFAs participate in multiple processes and their interaction with different receptor systems that are expressed in immune cells [47,48] and, accumulating experimental da-ta analyzed the immune-inflammatory cellular downstream effects of SCFAs. using ei-ther purchased-SCFAs bound to bound to bovine serum albumin (BSA) or SCFAs from direct fermentation of commensals….”.

Reference:

[47] Garrett, W.S. Immune recognition of microbial metabolites. Nat. Rev. Immunol. 2020, 20, 91–92, doi:10.1038/s41577-019-0252-2.

[48] Minihane, A.M.; Vinoy, S.; Russell, W.R.; Baka, A.; Roche, H.M.; Tuohy, K.M.; Teeling, J.L.; Blaak, E.E.; Fenech, M.; Vauzour, D.; et al. Low-grade inflammation, diet composition and health: Current research evidence and its translation. Br. J. Nutr. 2015, 114, 999–1012, doi:10.1017/S0007114515002093.

(Lines 160-166, Page 4):

“…By interacting with these receptors, in one study, butyrate-BSA inhibits reactive oxy-gen species production in neutrophils, it decreases the production of inflammatory cytokines (including monocyte chemoattractant protein-1 (MCP-1) and interleukin-6 (IL-6)) [51]) and it down-regulates the pro-inflammatory switching of macrophages [51]. Similarly, in an independent study, butyrate directly produced by gut commensals promotes the anti-inflammatory activity of regulatory T cells (Tregs) [52] (Figure 2).”.

References:

[51]     Ohira, H.; Fujioka, Y.; Katagiri, C.; Mamoto, R.; Aoyama-Ishikawa, M.; Amako, K.; Izumi, Y.; Nishiumi, S.; Yoshida, M.; Usami, M.; et al. Butyrate attenuates inflammation and lipolysis generated by the interaction of adipocytes and macrophages. J. Atheroscler. Thromb. 2013, 20, 425–442, doi:10.5551/jat.15065.

[52]     Arpaia, N.; Campbell, C.; Fan, X.; Dikiy, S.; Liu, H.; Cross, J.R.; Pfeffer, K.; Coffer, P.J.; Rudensky, A.Y.; Donald, B.; et al. Metabolites produced by commensal bacteria promote peripheral regolatory T cell generation. Nature 2013, 504, 451–455, doi:10.1038/nature12726.Metabolites.

(Lines 182-189, Page 5):

“By interacting with FFAR2, acetate-BSA promotes the phagocytic activity of macrophages, inhibits LPS-induced secretion of inflammatory cytokines by mononuclear cells [53], and triggers the production of oxygen free radicals at the sites of inflammation. Furthermore, mice fed with fibers-enriched diet and with acetic acid in water ad libitum were protected against C. difficile infection, due to interaction of gut commensals-derived acetate with FFAR2 that promotes the switching of the NOD-like receptor protein 3 (NLRP3)-inflammasome [54] (a central pathway in immune-metabolic derangements [55] and atherosclerosis [56]) (Figure 2).”.

References:

[53]     Masui, R.; Sasaki, M.; Funaki, Y.; Ogasawara, N.; Mizuno, M.; Iida, A.; Izawa, S.; Kondo, Y.; Ito, Y.; Tamura, Y.; et al. G protein-coupled receptor 43 moderates gut inflammation through cytokine regulation from mononuclear cells. Inflamm. Bowel Dis. 2013, 19, 2848–2856, doi:10.1097/01.MIB.0000435444.14860.ea.

[54]     Fachi, J.L.; Sécca, C.; Rodrigues, P.B.; Mato, F.C.P. de; Di Luccia, B.; Felipe, J. de S.; Pral, L.P.; Rungue, M.; Rocha, V. de M.; Sato, F.T.; et al. Acetate coordinates neutrophil and ILC3 responses against C. difficile through FFAR2. J. Exp. Med. 2020, 217, e20190489, doi:10.1084/jem.20190489.

[55]     Strowig, T.; Henao-Mejia, J.; Elinav, E.; Flavell, R. Inflammasomes in health and disease. Nature 2012, 481, 278–286, doi:10.1038/nature10759.

[56]     Baragetti, A.; Catapano, A.L.; Magni, P. Multifactorial activation of nlrp3 inflammasome: Relevance for a precision approach to atherosclerotic cardiovascular risk and disease. Int. J. Mol. Sci. 2020, 21, 1–13, doi:10.3390/ijms21124459.

(Line 195, Page 5):

“..With the same mechanism, butyrate (administered in mice in drinking water ad libitum) down-regulates LPS-induced..”

(Lines 205-206, Page 5):

“By contrast to butyrate and propionate, intra-gastric infusion of acetate promotes glu-cose intolerance…”.

 Part 2.2 Medium and long chain fatty acids

Authors reported: “In general, however, it is widely accepted that SFAs have pro-inflammatory effects, while MUFA’s and PUFA’s are anti-inflammatory”. I would suggest toning down that statement. Despite some evidence for pro-inflammatory effects of SFA, literature is not always consistent. Also, among PUFA’s, some members of n-6 family have been clearly identified as pro-inflammatory mediators (such as LA, ARA and eicosanoids). Thus, this paragraph seems to suffer from a lack of data on the effect of PUFA n-6 while the role of (various) SFA as well as DHA and EPA, are well described.

We acknowledge this gap and we have now amended this part by including this information as follows (Lines 223-225, Page: 6):

 “...The scenario however is complex, since contrasting data from literature impede to clearly discriminate pro- or anti-inflammatory properties of medium chain fatty acids.”

(Lines 251-262, Page 6):

“As compared to the n-3 series, n-6 PUFAs clearly demonstrated pro-inflammatory ef-fects. Arachidonic acid (ARA) (a 20 carbons, n6 PUFA that can be found only in ani-mal-derived foods) promotes oxidative metabolism [79] and its stimulate the release of IL-6 and TNF- by macrophages, through the production of two downstream products of ARA, the Leukotriene B4 (LTB4) and Prostaglandin E2 (PGE2 [80]). Also, further in vitro experiments postulated that ARA is able to activate the Janus-Kinase pathway in macrophages, inducing cell cycle arrest [81]. By contrast to this pro-inflammatory vi-sion, ARA might also support the efficiency of neutrophils in sites of inflammation [82] (Figure 2). In fact neutrophils, once activated at the site of inflammation through the Granulocyte-Colony Stimulating Factor (GM-CSF), increase the uptake of ARA contained in triglycerides by fatty acid transport protein 2 (FATP2) and convert ARA to PGE2 that promotes the suppression of CD8+ cytotoxic T cells [83].”.

References:

[79]     Adam, A.-C.; Lie, K.K.; Moren, M.; Skjærven, K.H. High dietary arachidonic acid levels induce changes in complex lipids and immune-related eicosanoids and increase levels of oxidised metabolites in zebrafish ( Danio rerio ). Br. J. Nutr. 2017, 117, 1075–1085, doi:10.1017/S0007114517000903.

[80]     Holladay, C.S.; Wright, R.M.; Spangelo, B.L. Arachidonic acid stimulates interleukin-6 release from rat peritoneal macrophages in vitro: Evidence for a prostacyclin-dependent mechanism. Prostaglandins, Leukot. Essent. Fat. Acids 1993, 49, 915–922, doi:10.1016/0952-3278(93)90176-W.

[81]     Shen, Z.; Ma, Y.; Ji, Z.; Hao, Y.; Yan, X.; Zhong, Y.; Tang, X.; Ren, W. Arachidonic acid induces macrophage cell cycle arrest through the JNK signaling pathway. Lipids Health Dis. 2018, 17, 26, doi:10.1186/s12944-018-0673-0.

[82]     Wellenstein, M.D.; de Visser, K.E. Fatty Acids Corrupt Neutrophils in Cancer. Cancer Cell 2019, 35, 827–829, doi:10.1016/j.ccell.2019.05.007.

[83]     Veglia, F.; Tyurin, V.A.; Blasi, M.; De Leo, A.; Kossenkov, A. V.; Donthireddy, L.; To, T.K.J.; Schug, Z.; Basu, S.; Wang, F.; et al. Fatty acid transport protein 2 reprograms neutrophils in cancer. Nature 2019, 569, 73–78, doi:10.1038/s41586-019-1118-2.

Accordingly, we have also included this information in Figure 2:

[THE FIGURE CAN BE VISUALIZED IN THE ATTACHMENT]

Part 2.3 Dietary cholesterol

This part deals with the atherogenic potential of lipoprotein particle (VLDL and oxLDL). VLDL are triglyceride-rich lipoprotein, and LDL (and oxLDL) carry endogenous cholesterol rather than dietary cholesterol. Although the authors define both exogenous and endogenous pathways, I would suggest modifying the title of this 2.3 part to avoid misunderstanding and confusion. Numerous data recently reported that there is no appreciable relationship between consumption of dietary cholesterol and serum cholesterol.  And in accordance with sciences-based evidence, the American Heart Association eliminated specific dietary cholesterol target in updated guidelines (2013).

We agree with the reviewer on these important aspects. We have now implemented a critical discussion on the effect of dietary cholesterol, including the information suggest by the reviewer.

Accordingly, we firstly revised the title of this part as follows:

The contribution of dietary cholesterol as compared to serum cholesterol

Also, we have revised the text as follows (Lines 321-331, Page 8):

“..Despite the average content of western style meals is estimated between 20 and 40 grams of total fats/meal and three-four meals/day are typically consumed [101,102], given the actual content of cholesterol in the majority of foods (by 4 to 700 milligrams per quantity of food consumed [103,104]), it appears clear that dietary source of choles-terol is minor as compared to that re-cycled through this complex metabolic system. Historical data actually demonstrated that there is a poor effect the physiological con-sumption of cholesterol by diet and changes in serum cholesterol [105]. In line with this, both the American Heart Association Guidelines of 2013 [106] and the more re-cent indications of the European Society of Cardiology (ESC) and European Athero-sclerosis Society (EAS [5]) toned down the magnitude of the effect and the level of evi-dence of reducing dietary cholesterol intake for CVD prevention.”.

References:

[5]       Mach, F.; Baigent, C.; Catapano, A.L.; Koskinas, K.C.; Casula, M.; Badimon, L.; Chapman, M.J.; De Backer, G.G.; Delgado, V.; Ference, B.A.; et al. 2019 ESC/EAS Guidelines for the management of dyslipidaemias: Lipid modification to reduce cardiovascular risk. Eur. Heart J. 2020, 41, 111–188.

[101]   Lopez-Miranda, J.; Williams, C.; Lairon, D. Dietary, physiological, genetic and pathological influences on postprandial lipid metabolism. Br. J. Nutr. 2007, 98, 458–473, doi:10.1017/S000711450774268X.

[102]   Sharrett, A.R.; Heiss, G.; Chambless, L.E.; Boerwinkle, E.; Coady, S.A.; Folsom, A.R.; Patsch, W. Metabolic and lifestyle determinants of postprandial lipemia differ from those of fasting triglycerides the Atherosclerosis Risk in Communities (ARIC) study. Arterioscler. Thromb. Vasc. Biol. 2001, 21, 275–281, doi:10.1161/01.ATV.21.2.275.

[103]   U.S. DEPARTMENT OF AGRICULTURE. FoodData Central Available online: https://fdc.nal.usda.gov/.

[104]   European Instutute of Oncology Food Composition database for Epidemiological Studies in Italy.

[105]   Berger, S.; Raman, G.; Vishwanathan, R.; Jacques, P.F.; Johnson, E.J. Dietary cholesterol and cardiovascular disease: a systematic review and meta-analysis. Am. J. Clin. Nutr. 2015, 102, 276–294, doi:10.3945/ajcn.114.100305.

[106] Eckel, R.H.; Jakicic, J.M.; Ard, J.D.; de Jesus, J.M.; Miller, N.H.; Hubbard, V.S.; Lee, I.-M.; Lichtenstein, A.H.; Loria, C.M.; Millen, B.E.; et al. 2013 AHA/ACC Guideline on Lifestyle Management to Reduce Cardiovascular Risk. Circulation 2014, 129, doi:10.1161/01.cir.0000437740.48606.d1.

(Lines 332-335, Page 8):

“Notwithstanding, by virtue of their diameter and because significantly smaller than chylomicrons, VLDL (30-70 nm) can enter the intima and it is plausible that die-tary cholesterol, accumulating over time, can exert pro-inflammatory effects in the vasculature (Figure 2).”.

(Lines 350-351, Page 8):

“Finally few, but robust experimental evidence suggests intra-cellular effects of cholesterol in promoting…”.

Part 3. 

Some statement may need clarification or little changes.

Page 9. The authors report “Among the series of SFAs, the increased consumption of total trans-fats has been historically associated with an increased risk of CHD…” : The link between SFAs and trans-fat here is not clear, as trans-fats are part of the unsaturated fatty acid family.

We apologize for the typo that occurred during the revision of the manuscript before submission. We have now clarified the sentence, removing the sentence “Among the series of SFAs…”.

The paragraph now starts with discussing evidence regarding trans-fats, separately from the previous paragraph, focused on SFAs.

Page 9 [… In fact, a higher content of ruminant-trans fat (rTFA), which are naturally found in foods, in plasma phospholipids was associated with reduced CVD risk factors …]. I would suggest being more accurate, as only very few foods are source of such ruminant TFAs: [In fact, a higher content of ruminant-trans fats (rTFA), which are naturally found in dairy foods and meat, in plasma phospholipids was associated with reduced CVD risk factors…]

We thank the reviewer for this suggestion which has been now implemented in the text as follows:

(Lines 412-413, Page 10):

“…a higher plasma phospholipids content of ruminant-trans fatty acid (rTFA), which are naturally found in dairy foods and meats,…”.

(Lines 423-424, Page 10):

Page 9 […Another example of this hypothesis is that the increased intake of SFAs from non-processed dairy products, despite not associated with CVD risk, correlates with…]. I would suggest modifying the sentence as follow: [… Another example of this hypothesis is that the increased intake of SFAs from whole fat dairy products, despite not associated with CVD risk, correlates with…].

Since we discussed on SFAs in the initial paragraph and we introduce the “matrix effect” hypothesis in this sentence, we have now revised the passage accordingly:

(Lines 423-424, Page 10):

 “..This hypothesis might be also applied to previously discussed SFAs from non-processed dairy products whose consumption, despite not associated with CVD risk [11],…”.

Reference:

[11]     Astrup, A.; Magkos, F.; Bier, D.M.; Brenna, J.T.; de Oliveira Otto, M.C.; Hill, J.O.; King, J.C.; Mente, A.; Ordovas, J.M.; Volek, J.S.; et al. Saturated Fats and Health: A Reassessment and Proposal for Food-Based Recommendations: JACC State-of-the-Art Review. J. Am. Coll. Cardiol. 2020, 76, 844–857.

Page 10. [ Dietary cholesterol presents in multiple processed and non-processed food patterns, including eggs, butter, beef, cheese butter and shrimps, deserves a separate discussion]. “Cheese butter” is an unknown product among dairy foods. This sentence should be: [ Dietary cholesterol presents in multiple processed and non-processed food patterns, including eggs, butter, beef, cheese and shrimps, deserves a separate discussion].

We apologize for the typo that occurred during the revision of the manuscript before submission. We revised accordingly.

Page 11. Table 2 (interventional studies). It is not clear how the authors resume the contrasting results from literature. For example, the effect of cholesterol is resumed as: “pro-inflammatory” and “increase CVD risk markers”, while both studies [167 and 168] from Blesso et al. showed that an increase in dietary cholesterol resulted in reduction of inflammatory markers. And while the 2 other studies referenced [169 and 170] reported that CRP level did not increase in patients with high cholesterol high fat diet compared to low cholesterol low fat diet (although affecting HDL composition and functionality).

We apology the reviewer for the improper indication in the Table.

We have now re-scheduled both Table 1 and Table 2 indicating, close to each symbol, the relative reference. In this way:

  1. For Table 2 “↑”: Indicates data showing a positive association between the dietary fat intake and the outcome (either inflammatory markers or CVD risk factors); “↓” Indicates data showing a positive association between the dietary fat intake and the outcome (either inflammatory markers or CVD risk factors). “↔” Indicates that there are missing or contrasting data regarding association between the dietary fat intake and the outcome (either inflammatory markers or CVD risk factors).
  2. For Table 1: “↑”: Indicates data showing a positive association between the dietary intervention and the outcome (either inflammatory markers or CVD risk factors); “↓” Indicates data showing a positive association between the dietary intervention and the outcome (either inflammatory markers or CVD risk factors). “↔” Indicates that there are missing or contrasting data regarding association between the dietary intervention and the outcome (either inflammatory markers or CVD risk factors).

These information has been included in the caption of the Tables as follows:

[THE TABLES CAN BE VISUALIZED IN THE ATTACHMENT]

Reviewer 2 Report

A well written review in regards to the target area of diet and CVD prevention. I like the way the authors took the approach connecting all the main dots starting with dietary link, gut bacteria and clinical study data. I only have few comments on the flow of the review. More specific comments are below:

Line 24 is missing conjuntion 

Line 107 to 110 where the authors say  "Beside the production of SCFAs, other gut microbial species express enzymatic systems that metabolize dietary lipids into inflammatory molecules; among them, trimethylamine (TMA) lyase, an enzyme that converts choline, a by-product of dietary phosphatidycholine, into TMA, is peculiarly expressed by Eggerthella lenta and Eggerthella timonensis" needs to have better flow

Line 111:  Gram- commensals shove be "-ve"

Author Response

REVIEWER#2

A well written review in regards to the target area of diet and CVD prevention. I like the way the authors took the approach connecting all the main dots starting with dietary link, gut bacteria and clinical study data.

Dear Reviewer,

We are grateful for your very positive evaluation of our work.

Please, find below replies to your questions and indication on revision that we applied (in red) in the text:

I only have few comments on the flow of the review. More specific comments are below:

Line 24 is missing conjuntion 

We have revised the sentence in the abstract (lines 23-24, page 1) as follows:

“…. Interventional trials, although providing stronger potential for causal inference, are typically small sample-sized, they have short follow-up, noncompliance, high attrition rates. Besides…”.

Line 107 to 110 where the authors say  "Beside the production of SCFAs, other gut microbial species express enzymatic systems that metabolize dietary lipids into inflammatory molecules; among them, trimethylamine (TMA) lyase, an enzyme that converts choline, a by-product of dietary phosphatidycholine, into TMA, is peculiarly expressed by Eggerthella lenta and Eggerthella timonensis" needs to have better flow

We thank the reviewer for indicating this passage. We have revised as follows (lines 115-117, page 3).

“Among them, trimethylamine (TMA) lyase, an enzyme that converts dietary phospha-tidycholine and choline into TMA, is peculiarly expressed by Eggerthella lenta and Eg-gerthella timonensis [35,36]).”.

References:

[35]     Wright, A.T. Gut commensals make choline too. Nat. Microbiol. 2019, 4, 4–5, doi:10.1038/s41564-018-0325-1.

[36]     Koeth, R.A.; Lam-Galvez, B.R.; Kirsop, J.; Wang, Z.; Levison, B.S.; Gu, X.; Copeland, M.F.; Bartlett, D.; Cody, D.B.; Dai, H.J.; et al. L-Carnitine in omnivorous diets induces an atherogenic gut microbial pathway in humans. J. Clin. Invest. 2019, 129, 373–387, doi:10.1172/JCI94601.

Line 111:  Gram- commensals shove be "-ve"

We changed accordingly (lines 117-118, Page 3):

“…Finally, gut resident Gram-negative commensals (e.g.: Escherichia coli, Salmonella minnesota, Salmonella typhimurium) synthetize…”.
